# Fall-related measures in elderly individuals and Parkinson's disease subjects

Justyna Michalska[ID][1][☯]*, Anna Kamieniarz[ID][1][☯], Anna Brachman[ID][1][☯], Wojciech Marszałek[1][☯], Joanna Cholewa[ID][2][‡], Grzegorz Juras[ID][1][‡], Kajetan J. Słomka[1][‡]

1 Institute of Sport Sciences, The Jerzy Kukuczka Academy of Physical Education, Katowice, Poland,
2 Institute of Physiotherapy and Health Sciences, The Jerzy Kukuczka Academy of Physical Education, Katowice, Poland

☯ These authors contributed equally to this work.
‡ These authors also contributed equally to this work.
* j.michalska@awf.katowice.pl

**Data Availability Statement:** All relevant data are within the manuscript and its Supporting Information files.

## Abstract

Falls pose a serious problem in elderly and clinical populations. Most often, they lead to a loss of mobility and independence. They might also be an indirect cause of death. The aim of this study was to determine an objective predictor of the fear of falling and falls in elderly subjects (ESs) and Parkinson's disease (PD) subjects. Thirty-two ESs were examined in this study, of whom sixteen were diagnosed with PD. The testing procedures comprised force plate measurements (limit of stability test–LOS test) and clinical tests (Berg Balance Scale, Functional Reach Test, Timed Up and Go test, Tinetti test). The Falls Efficacy Scale International (FES-I) was used to evaluate the fear of falling. The range of the maximum forward lean was normalized to the length from the ankle joint to the head of the first metatarsal bone and was named the functional forward stability indicator (FFSI). The FFSI, derived from the LOS test, allowed us to demonstrate the real deficit in functional stability and individual safety margins. Moreover, the FFSI was highly correlated with the FES-I score and almost all clinical test results in elderly subjects (r>0,6; p<0.05). In PD subjects, the FFSI was poorly correlated with the fear of falling, the BBS score and the FR distance; however, a high correlation with the Tinetii test (r>0,6, p<0.05) was noted. The PD subjects presented a different balance strategy when close to their stability limits, which was also reflected in the lower values of sample entropy (t = (-2.40); p<0.05; d = 0.87). The FFSI might be a good predictor of the fear of falling in the group of elderly people. Additionally, the FFSI allows us to show real balance deficits both in PD subjects and in their healthy peers without the need for a reference group and norms. In conclusion, it is postulated that the popular clinical assessments of postural balance in PD subjects should be accompanied by reliable posturography measurements.

## Introduction

Unintentional falls among people over 65 yr. of age are an increasing problem and are the leading cause of injury-related deaths [1]. According to the WHO, between 2015 and 2050, the

**Funding:** The study was supported by the National Center for Research and Development Grant under the program STRATEG MED III within the "VB-Clinic" project no. STRATEGMED3/306011/1/NCBR/2017.

**Competing interests:** The authors have declared that no competing interests exist.

proportion of the world's population over 60 years old will nearly double, from 12% to 22%. Currently, in developed countries, the percentage of elderly subjects (ESs) is more than 20%. ESs experience falls due to significant deterioration of balance ability associated with a decrease in muscle strength, sensory function and widespread degeneration in the central nervous system [2].

Moreover, with age, accompanying neurological diseases may appear. Parkinson's disease (PD) is the second most common neurodegenerative disorder [3]. This chronic and progressive disease leads to rigidity, tremors, bradykinesia and thus impaired balance [4]. Therefore, PD subjects are more exposed to the risk of falls than their peers [5].

The fear of falling (FOF) could appear well before and/or after an incident of a fall which creates a vicious cycle [6]. Concomitant psychological symptoms of falls are the FOF, which is common among older adults regardless of whether they have sustained a fall. The FOF might also be a proper reaction to certain situations, leading elderly subject to be cautious, and can contribute to fall prevention through careful choice of movement activity [7]. Nevertheless, ESs increasingly cease physical activity because of a FOF [6].

The FOF is assessed with many self-reported questionnaires, among which the Falls Efficacy Scale International (FES-I) test is widely used [8]. It has become a useful instrument for predicting the risk of falls in elderly subjects due to its excellent reliability and validity across different populations [9]. The questionnaire refers to most basic, everyday activities. Only people with a FOF may have difficulty performing these things.

Most daily activities require keeping the center of gravity (COG) within the confines of the base of support (BOS). Therefore, falls are quite frequent during everyday activities, especially when the COG is voluntarily shifted near or outside the limit of stability (LOS) [10]. The LOS is depicted by the stability boundary and anatomically was assumed to be the foot envelope. The COG usually travels along a much smaller area of the BOS [11]. It is suggested that the anterior mechanical stability boundary is located on the line connecting the extreme points located in the sagittal plane at the height of the first heads of the metatarsal bones [12]. Moreover, from a biomechanical point of view, the rotation axis is located slightly further than the area of the first metatarsophalangeal joint [13]. Crossing this mechanical limit inevitably involves changing the standing strategy (heel rise or toes co-contraction) or imposing gait initiation.

In older adults over 65 yr, 60% of falls occur in a forward direction [14]. Additionally, Youn et al. [15] noticed that more than 70% of PD subjects had a history of falls in the forward direction. Therefore, in the present study, we focused on exploring the anterior stability boundary. The most frequent measure used to identify the stability boundary is the limit of stability (LOS) test [16]. During this test, the detachment of the heel from the ground is prohibited. The calculation of the maximum forward lean range with respect to the mechanical (anatomical) stability boundary should allow subjects to reach the boundary that is in close proximity to the real safety margin. It is crucial for appropriate assessment of balance instability [12].

Clinical assessment of the risk of falling is based on the Berg Balance Scale (BBS), the Tinetti test or the Timed Up and Go (TUG) test. These tests aim to evaluate the integration of results of the systems involved in postural stability. Unfortunately, clinical tests do not facilitate the understanding of the etiology of fall-related concerns [17]. Clinical tests are often based on a qualitative assessment (BBS, Tinetti test). Despite the clear assessment instructions, the researcher may influence the test results. We postulate more specific and reliable tools to be used for the diagnosis and comprehension of complex postural control processes [18]. With an inefficient postural strategy (ankle and hip strategy) in elderly subjects, falls usually occur during movements where the COG is shifted near or outside the BOS [19]. It seems that the LOS test is the most appropriate measurement. This is consistent with Johansson et al. [20]

report which shows that postural sway in relation to assisted stability limits appears to be a valid predictor of incident falls. Additionally, posturography allows one to assess the difficulty of the motor task by analyzing COP regularity. The regularity of COP trajectories is quantified by sample entropy (SampEn). This was found to be positively related to the degree of attention invested in postural control [21]. Higher values of SampEn indicate more irregularity in the COP signals. The basic assumption is that the automatic control processes increase the entropy of the signal while volitional control decreases the signal [22]. In a certain sense, an analysis of SampEn should allow one to determine for which study group (ESs or PD subjects) the LOS test was more challenging.

The aim of this study was to determine an objective predictor of the fear of falling and falls in ESs and patients with neurodegenerative disorders. For this purpose, the clinical and laboratory tests were analyzed. Depending on the study groups, the proposed objective measure could replace or support clinical assessment. Posturographic examination can also register early and nonvisible postural control changes, so we assumed that this objective assessment might replace clinical tests among ESs. We also hypothesized that the identification of the real functional limit of support with the LOS test would allow us to assess not only the fear of falling but also the risk of falls. We assumed that PD subjects present different balance strategies when their COG is located near their stability boundary. The analysis of the SampEn should allow us to identify these changes.

## Methods

### Subjects

Sixteen PD subjects and sixteen anthropometrically matched healthy controls voluntarily participated in the study (Table 1). PD subjects were recruited from the Local Association of Parkinson's disease. The research took place at the Academy of Physical Education in Katowice. Participants were males and females. A preliminary intragroup comparison with respect to the analyzed variables revealed that sex did not differentiate the groups. Based on a medical

**Table 1. Characteristics of subjects.**

|  | Elderly subjects | Parkinson's disease subjects | test values |
|---|---|---|---|
| Sample size | 8M/8F | 9M/7F | - |
| Age [years] | 67 (65–69) | 65 (65–69.5) | t = -1.35 |
| Height [cm] | 163.8 (160.8–166.8) | 167 (165.2–174.2) | t = 0.70 |
| Weight [kg] | 74.8 (68.07–81.6) | 77.9 (71.6–84.2) | t = 1.95 |
| FES-I (score) | 11.4 (8.4–14.27) | 12.3 (9.3–15.2) | U = 121.5 |
| Low concern | n = 5 (31.25%) | n = 7 (43.75%) | - |
| Moderate concern | n = 7 (43.75%) | n = 3 (18.75%) | - |
| High concern | n = 4 (25%) | n = 6 (37.5) | - |
| Fellers | n = 3 (18.75%) | n = 6 (37.5%) | - |
| FFSI (%) | 82.78 (77.7–87.9) | 81.00 (74.2–87.9) | U = 128 |
| BBS (score) | 54.4 (53.3–55.5) | 47.6 (41.4–55.4) | U = 73.5 |
| FR (cm) | 27 (24.6–29.3) | 22 (17.6–28) | U = 104 |
| Tinetti test (score) | 27.5 (27.2–27.8) | 23 (19.9–26.6) | U = 96 |
| TUG (s) | 7.14 (6.5–7.7) | 10.28 (8.1–13.5) | U = 72* |

Abbreviations: variables presented by mean values and 95% confidence intervals (CI), F, female; M: male FES-I, Falls Efficacy Scale International; FFSI, Functional Forward Stability Indicator; BBS, Berg Balance Scale; FR, Functional Reach; TUG, Timed Up and Go

* indicates a statistically significant correlation $p < 0.05$

interview, the general inclusion criteria for PD subjects were age $\geq$ 65 years, identified and diagnosed Parkinson's disease, and stage III disease according to the Hoehn and Yahr scale. The exclusion criteria were as follows: no consent to participate in the research; dementia or cognitive impairment (MMSE score below 26 points); and neuromuscular, vestibular or orthopedic disorders. PD subjects were tested during the "ON period" of their usual antiparkinsonian medication (1 h after taking their usual dose of medication). The subjects gave informed written consent for voluntary participation in the study. The research was approved by the Institutional Ethics Committee of the Medical University of Warsaw (number KB/28/2014).

## Apparatuses and procedures

Functional stability was evaluated with a force platform (AMTI, Accugait, Watertown, MA, USA). The anthropometrics of the foot were measured with a spreading caliper. The Falls Efficacy Scale International (FES-I) was used to evaluate the fear of falling. It includes 7 activities with four possible answers: not at all concerned (one point), somewhat concerned (two points), fairly concerned (three points) and very concerned (four points). The highest score (very concerned about falling) is, therefore, 28, and the minimum (no concern about falling) score is 7. Additionally, subjects were asked about their history of falls, which was defined as "*unintentionally coming to rest on the ground or a lower level*" during the preceding year [23]. Information about the cause and direction of falls as well as possible injury from falls was also collected. The LOS test was used to evaluate functional stability and was conducted as described in the study by Juras et al. [24]. This test consists of three distinct phases. The 1st phase includes quiet standing (QS) and lasts 10 seconds. During the 2nd phase, after an auditory signal, subjects lean forward as quickly and far as they can. The 3rd phase is associated with maintaining the inclined position. The fixation point is placed 3 m away from the participants on the wall in front of them at eye level. The measurement was interrupted and repeated if subjects lifted their heels. The LOS test lasted 30 seconds and was repeated three times. The average values of posturographic variables of three repetitions were taken for further analysis.

The testing procedures also involved clinical tests widely applied for balance assessment in ESs and PD subjects, such as the BBS, Tinetti test, FR test and TUG test. In the first three tests, a higher score indicates better balance ability. A shorter time in the TUG test suggests a good result.

## Data processing

Raw platform data were processed offline using MATLAB (MathWorks Inc., Natick, MA, USA, v. r2017b). A low-pass 4-order Butterworth filter with a sampling frequency of 7 Hz was used for forces (Fx, Fy, Fz) and moments (Mx, My, Mz), which were later used to calculate the center of foot pressure (COP). Further COP analysis was based on the anatomical foot characteristics. The platform sampling frequency was 100 Hz. The following variables of COP displacement were calculated for the 1st and 3rd phases of the LOS test: SampEn in an anterior-posterior (AP) and medio-lateral (ML) plane [25]. In the 2nd phase of the LOS test, the range of the maximum forward lean was normalized to the length from the ankle joint to the head of the first metatarsal bone and was named the functional forward stability indicator (FFSI). The calculation of the forward functional stability indicator [%] (FFSI) was conducted as follows in Eq (1):

$$FFSI\ [\%] = \frac{NMVE}{FFL}\ x\ 100 \tag{1}$$

*Abbreviations NMVE–normalized maximal voluntary COP excursion range calculated from the position of the medial malleolus to the average COP in the third phase of the LOS test, corresponding to maintenance in the inclined position [cm],*

*FFL–the length of the forefoot [cm].*

### Statistical analysis

The basic parameters of the descriptive statistics were calculated and analyzed. The Shapiro-Wilk test was used to check the data for normal distribution. The parametric and nonparametric tests were used. An independent-sample t-test was conducted to compare the anthropometric variables FFSI and SampEn from ESs and PD subjects. The effect sizes for the main and interaction effects are reported as Cohen's d. The Mann-Whitney U test was conducted to compare values of FES-I, BBS, Tinetti, FR and TUG test from ESs and PD subjects. Spearman's rank test was used to evaluate the associations among the FFSI and FES-I, clinical test results, and the number of falls. The levels of correlation were considered weak positive (0–0.3), weak negative (-0.3–0), moderate positive (0.3–0.7), moderate negative (-0.3 –(-0.7)), high positive (0.7–1.0), and high negative (-0.7 –(-1.0)) [26]. Intraclass correlation coefficients (ICCs) were calculated for the LOS test (FFSI, SampEn phase 1, SampEn phase 3). The levels of reliability were considered poor (ICC < 0.50), moderate ($0.50 \leq$ ICC < 0.75), good ($0.75 \leq$ ICC < 0.90), and excellent (ICC $\geq$ 0.90), according to Portney et al. [27]. The LOS test achieved excellent reliability with 3 repetitions. The level of significance was set at p $\leq$. 0.05. The Bonferroni correction was used to decrease risk of a type I error during making multiple statistical tests. All calculations were carried out using STATISTICA v.13.1 (StatSoft, Inc., USA).

## Results

### The analysis of the limit of stability of the COP characteristics between the groups

There were no significant differences between ESs and PD subjects in the FFSI (t = (-0.44); p>0.05; d = 0.16). However, ESs leaned further than PD. Additionally, during the 1st phase of the LOS test, there were no significant differences in the value of SampEn between the two groups (t = (-1.99); p>0.05; d = 0.73). Significantly lower values of SampEn in the 3rd phase of the LOS test were observed in PD subjects than in ESs (t = (-2.40); p<0.05; d = 0.87) (Fig 1).

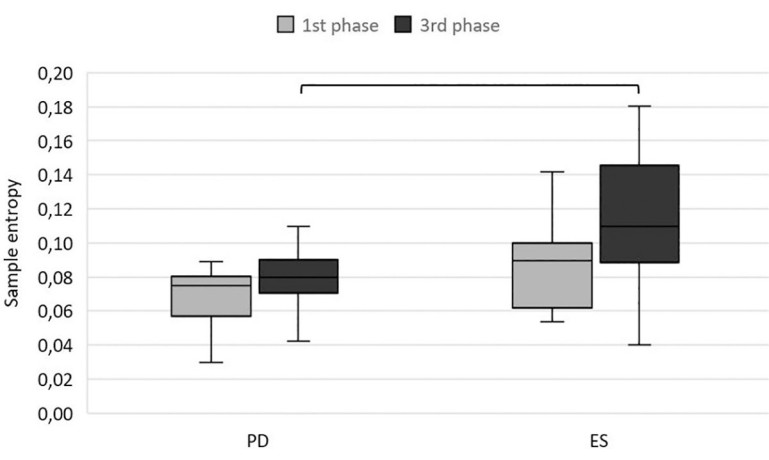

**Fig 1. The median value of sample entropy in the anterior-posterior direction (minimum and maximum marked as error bars) in the first and third phases of the limit of stability test.** Abbreviations: ESs, elderly subjects; PD, Parkinson's disease subjects.

**Table 2. Spearman's correlation coefficient between FFSI and balance clinical test in ESs and PD subjects.**

|  | Group | BBS | FR | Tinetti | TUG | FES-I | NoF |
|---|---|---|---|---|---|---|---|
| FFSI [%] | ES | 0,634* | 0,674** | 0,759** | -0,874** | -0,831** | -0,229 |
|  | PD | 0,352 | 0,471 | 0,698** | -0,626* | -0,480 | -0,604* |

Abbreviations: ESs, elderly subjects; PD, Parkinson's disease subjects; FFSI, functional forward stability indicator; FES-I, Falls Efficacy Scale International; BBS, Berg Balance Scale; FR, Functional Reach; TUG, Timed Up and Go; NoF, number of falls

* indicates a statistically significant correlation p<0.05

** indicates a statistically significant correlation p<0.00833 (Bonferonni correction)

bold indicates a statistically significant correlation p<0.05.

### Correlation between the FFSI and the fear and risk of falling in elderly subjects

A significant positive correlation was observed between the FFSI and clinical test results, such as the FR test and Tinetti (p< 0.00833). There was also a significant negative correlation between the FFSI and the TUG test and FES-I scores (p< 0.00833). The numbers of falls did not show any correlations with maximum forward lean as assessed by the FFSI (p> 0.00833) (Table 2).

### Correlation between the FFSI and the fear and risk of falling in Parkinson's disease subjects

There was a significant positive correlation between the FFSI and Tinetti test scores (p< 0.00833). BBS, FR, TUG test and FES-I scores and number of falls did not show any correlations with maximum forward lean as assessed by the FFSI (p> 0.008333) (Table 2).

### Correlation between clinical tests and the fear and risk of falling in elderly subjects and PD subjects

There was only one significant negative correlation between the fear of falling and the TUG test scores in the ES group (p< 0.00625). The rest of the clinical test scores did not correlate with the FES-I or the number of falls. In contrast, in PD subjects, BBS, and Tinetti scores were significantly negatively correlated with the FES-I and TUG test was significantly positively correlated with the FES-I (p< 0.00625). Additionally Tinetti test scores showed a significant negative correlation with the number of falls (p< 0.00625). There were no correlations between BBS, FR and TUG test and the number of falls (p> 0.00625) (Table 3).

## Discussion

The main aim of this study was to determine an objective predictor of the fear of falling and falls in ESs and PD subjects. That is way, we have analyzed both the qualitative and quantitative balance diagnostic methods, indicating that the posturographic examination can both register early and nonvisible postural control changes. Additionally LOS test assesses safety margin, as an alert of fear of falling. First, we assumed that the identification of the real functional limit of support with the LOS test would allow us to assess not only the fear of falling but also the risk of falls. Second, we hypothesized that depending on the study groups, the proposed objective measure could replace or support clinical assessment. Third, we supposed that PD subjects present different balance strategies when their COG is located near their stability boundary.

The FR test and the LOS test are the most frequent measures of functional balance and the safety margin, suggested by Clark et al. [28] to be complementary due to the differences in task

**Table 3. Spearman's correlation coefficients between among balance clinical test scores, the FES-I score, and the number of falls in elderly subjects and PD subjects.**

| Group | Balance clinical tests | FES-I | Number of falls |
|---|---|---|---|
| ES | BBS | -0,481 | -0,203 |
| | FR | -0,468 | -0,313 |
| | Tinettii | -0,421 | -0,199 |
| | TUG | 0,703** | 0,134 |
| PD | BBS | -0,698** | -0,404 |
| | FR | -0,554* | -0,346 |
| | Tinettii | -0,858** | -0,689** |
| | TUG | 0,723** | 0,582* |

Abbreviations: ESs, elderly subjects; PD, Parkinson's disease subjects; FES-I, Falls Efficacy Scale International; BBS, Berg Balance Scale; FR, Functional Reach; TUG, Timed Up and Go

* indicates a statistically significant correlation p<0.05

** indicates a statistically significant correlation p<0.00625 (Bonferonni correction)

constraints. Nevertheless, neither of them differentiated PD subjects from their healthy peers. Similar results were reported by Ryckewaert et al. [29] and Mancini et al. [30]. On the other hand, significant differences between PD subjects and their healthy peers were presented by Behrman et al. [31], where the authors observed a significantly lower range of FR distance in PD subjects in comparison to the control group. They have also found that FR test scores below 25 cm indicate a high risk of falls. Based on this we would qualify our PD subjects, with their functional reach distance of 22,8 cm, as the at-risk group, even though there were no significant differences between the study groups. The latter might suggest that the LOS test is less valuable than the FR test. However, it is postulated that the range of maximum forward lean calculated with respect to the anatomical stability boundary allows us to demonstrate the real deficit of functional stability [32], which we believe is very important information with high practical value.

ESs used 82.78% of their functional BOS, which is located between the head of the first metatarsal bone and the medial malleolus (ankle joint). Their safety margin equaled 17.22%, in comparison to 19% for PD subjects. So PD subjects used a smaller area of the functional BOS during the LOS test than ESs (Fig 2). A greater safety margin decreases the range of the maximum forward lean, thereby decreasing functional movement. So the existence of a safety margin might indicate deficits in balance. In our study higher values of safety margin (lower values od FFSI) correlated with clinical tests which assess risk of falls, which confirms our first research hypothesis (Table 2). According to Słomka et al. [32], a safety margin is located between the first metatarsal phalangeal joint and the COP location while keeping the body in a leaning position. Therefore, the lower range of the maximum forward lean increases the area of the safety margin. A greater safety margin is combined with greater postural instability [12]. It is obvious that appropriate rehabilitation can improve balance [33]. Therefore, purposeful therapy should be aimed at reducing the safety margin and allowing the relocation of the COP near the anatomical stability boundary during the LOS test. The FFSI bears clear information about functional stability deficits in the sagittal plane, and it also indicates a target value in the range of maximum forward COP excursion for patients in balance rehabilitation.

We also postulated that existence of a safety margin correlated not only with risk of falls but also with FOF. In the present study, the FFSI was highly correlated with the FES-I in the control group. The decreased range of maximum forward lean indicated an increased postural threat. In turn, there were no correlations between FR test scores and both the FOF and the

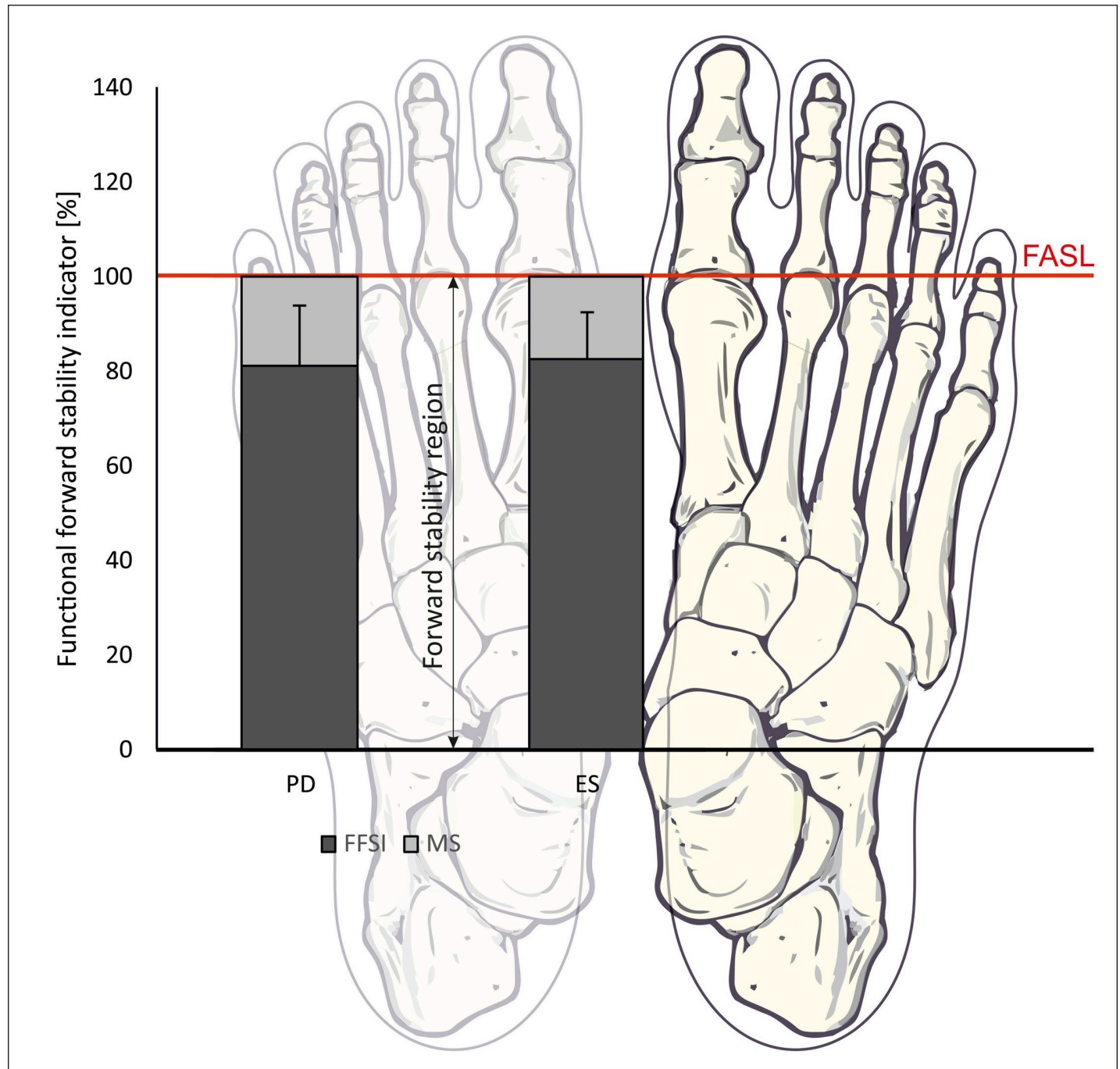

**Fig 2. Forward stability region.** Abbreviations: ESs, elderly subjects; PD, Parkinson's disease subjects; FFSI, functional forward stability indicator; FASL, forward anatomical stability limit; MS, margin safety.

history of falls in healthy ESs. The LOS test is more specific than the FR test because it stresses the use of the ankle strategy [34], which minimizes early and small perturbations of upright posture. Different movement strategies adopted in FR test may significantly affect the final result and include the rotation of the trunk and a combination of the ankle and hip strategy [35]. The LOS test is less prone to such strategy alterations during performance. We also

confirmed our hypothesis that the LOS test might replace FR test in the diagnosis of ESs, which leads to conclusion that among healthy ESs the computerized posturography should be considered a more appropriate measure for the early identification of balance deficits than subjective tests and FR test.

Other clinical balance test scores as well as FR test are poor predictors of the fear of falling in groups of healthy ESs. In the present study, only TUG test were correlated with FES-I scores. Neither of them were correlated with a history of falls. Landers et al. [36] showed which self-reported questionnaires are most clinically applicable to the prediction of falls in the elderly population. They also indicated that TUG test is the most predictive clinical test of the risk of falls. In general, clinical tests detect visible balance deficits. However, the identification of postural control changes that are invisible to the naked eye should be pursued. The FFSI allows the detection of the area of the safety margin, and it is correlated with the FOF in healthy ESs. Moreover, the FFSI is also highly correlated with clinical test scores, such as TUG and Tinetti test scores, which are known to be good predictors of the risk of falls [2]. We can conclude that, based on our results, it is not necessary to use the whole battery of clinical tests for balance assessment to achieve satisfactory diagnostic outcomes. Acquiring more information in less time is especially important when conducting diagnoses in specific populations and situations. We also confirmed our hypothesis that in group of ESs objective measures could replace clinical assessment.

The diagnosis of the PD subjects led to different observations. According to the Hoehn and Yahr scale, all PD subjects in this study were in stage III. Everyone had visible and persistent changes in postural control [3], which could be detected by both clinical tests and posturography measurements. All clinical test scores were highly correlated with the FOF (except for FR). Additionally, the Tinetti test was highly correlated with a history of falls. Nevertheless, the FFSI had a poor correlation with the FOF and with history of falls. Moreover, the FFSI was associated with only one clinical tests of the risk of falls. Our results validated our second research hypothesis; therefore we postulate that balance assessments of the PD population should be made both with clinical tests and with computerized posturography. Clinical tests also assess movements such as walking forward, quiet standing and sitting down, which are indicated by Robinovitch et al. [37] as the most risk of falling activities.

As shown, the differences between PD subjects and their healthy peers can be easily detected with TUG tests. TUG test is an incontestably appropriate measure to assess the dynamic balance among PD subjects, using the BBS for dynamic balance assessment is more questionable [38]. Some authors indicated that the BBS, when used to measure balance in patients with PD, may not completely assess the whole spectrum of postural control impairments typical for this disease [39]. Moreover, La Porta et al. [40] suggested that the BBS, even if modified by the new rating scale, may not be an effective tool for the measurement of early postural control disturbance in patients with PD.

Although there were no differences between PD subjects and the control group, it could be observed that the LOS test was more demanding for PD subjects. During the 3rd phase of the test, where subjects had to maintain an inclined position, PD subjects were characterized by lower values of SampEn. According to Richman and Moorman [25], low values of entropy show more regularity in the COP signal. Therefore, less chaotic excursions may be defined as a characteristic of an unsuccessful vigilant strategy to maintain balance. Furthermore, Donker et al. [41] showed that the amount of attention invested in keeping an upright posture is positively correlated with sway regularity. Based on these reports, PD subjects paid more attention to maintaining an inclined position than the control group (Fig 1). Therefore, we confirmed our hypothesis that the LOS test is a more challenging motor task for PD subjects and they presented different balance strategy. Reinert et al. [42] confirmed that the use of nonlinear

analysis methods such as SampleEn may improve the ability of the LOS test to identify differences in performance that are not currently revealed by traditional sway ranges.

This study has several limitations. A larger sample size would have been better as it would help to better generalize the results. Additionally, it is known that psychological factors, which had not been taken into account in this study, are associated with increased postural threat and could also affect the processing of postural control. All PD subjects were in stage III and they took regularly antiparkinsonian medication. It is known that pharmacological treatments may have an impact on postural control in PD subjects. Although we did our best to satisfy the standardization of procedures, we did not explore the problem of the impact of medication on postural control. Future research should be focused on the prospective risk of falls. It will be useful to create FFSI norms that can help diagnose the risk of falls.

## Conclusion

The present study highlights the importance of a quantitative approach for balance evaluation in ESs and elderly subjects with neurological disorders. The functional forward stability indicator is a good predictor of the fear of falling in the group of elderly people. Additionally, the FFSI allows us to show real balance deficits both in PD subjects and in their healthy peers without using an external reference group or created norms. A complete assessment of postural balance in PD subjects should be made both with posturography measurements and with clinical tests.

## Supporting information

**S1 Data.**
(CSV)

## Author Contributions

**Conceptualization:** Justyna Michalska, Anna Kamieniarz, Anna Brachman, Wojciech Marszałek, Joanna Cholewa, Kajetan J. Słomka.

**Data curation:** Justyna Michalska, Anna Kamieniarz, Wojciech Marszałek, Kajetan J. Słomka.

**Formal analysis:** Justyna Michalska.

**Funding acquisition:** Grzegorz Juras.

**Investigation:** Justyna Michalska, Anna Kamieniarz, Wojciech Marszałek, Joanna Cholewa.

**Methodology:** Justyna Michalska, Anna Kamieniarz, Anna Brachman, Kajetan J. Słomka.

**Project administration:** Justyna Michalska, Anna Kamieniarz, Anna Brachman, Grzegorz Juras.

**Resources:** Joanna Cholewa, Grzegorz Juras.

**Software:** Wojciech Marszałek.

**Supervision:** Justyna Michalska, Anna Brachman, Grzegorz Juras, Kajetan J. Słomka.

**Validation:** Justyna Michalska, Anna Kamieniarz, Wojciech Marszałek, Grzegorz Juras.

**Visualization:** Justyna Michalska, Anna Kamieniarz, Wojciech Marszałek, Kajetan J. Słomka.

**Writing – original draft:** Justyna Michalska, Kajetan J. Słomka.

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
