## [Decision Letter · Decision Letter 0]

30 Apr 2020

PONE-D-20-08972

Predictive measures of falls in elderly and Parkinson’s Disease

PLOS ONE

Dear Mrs Michalska,

Thank you for submitting your manuscript to PLOS ONE. After careful consideration, we feel that it has merit but does not fully meet PLOS ONE’s publication criteria as it currently stands. Therefore, we invite you to submit a revised version of the manuscript that addresses the points raised during the review process.

Two experts in the field have carefully evaluated the manuscript entitled, “Predictive measures of falls in elderly and Parkinson’s Disease”. Their comments are appended below.

Each reviewer pointed out the drawbacks which should be considered before publication. They both raised several detailed serious concerns through the manuscript. Although the first reviewer acknowledged the manuscript is worth for publication leaving various shortcomings, the second one, however, criticized English description is difficult to read through. Thus this reviewer advised that authors need receiving professional proof reading before submission.

This Academic Editor, myself, is sure your revision will strengthen your study and I am expecting to receive your reply to each comment and the necessary revision.

We would appreciate receiving your revised manuscript by Jun 14 2020 11:59PM. To enhance the reproducibility of your results, we recommend that if applicable you deposit your laboratory protocols in protocols.io, where a protocol can be assigned its own identifier (DOI) such that it can be cited independently in the future. For instructions see: http://journals.plos.org/plosone/s/submission-guidelines#loc-laboratory-protocols

We look forward to receiving your revised manuscript.

Kind regards,

Manabu Sakakibara, Ph.D.

Academic Editor

PLOS ONE

3. Please ensure that you refer to Figure 1 in your text as, if accepted, production will need this reference to link the reader to the figure.

4. We note you have included a table to which you do not refer in the text of your manuscript. Please ensure that you refer to Table 1 in your text; if accepted, production will need this reference to link the reader to the Table.

Reviewers' comments:

Reviewer's Responses to Questions

**Comments to the Author**

1. Is the manuscript technically sound, and do the data support the conclusions?

Reviewer #1: Yes

Reviewer #2: Partly

2. Has the statistical analysis been performed appropriately and rigorously? 

Reviewer #1: Yes

Reviewer #2: No

3. Have the authors made all data underlying the findings in their manuscript fully available?

Reviewer #1: Yes

Reviewer #2: Yes

4. Is the manuscript presented in an intelligible fashion and written in standard English?

Reviewer #1: No

Reviewer #2: No

5. Review Comments to the Author

Reviewer #1: Dear authors, Thank you for your submission and your contribution to the field. This study is interesting with some worthy outcomes.

I have attached a copy of your submission with some comments and suggestions. In summary, my main reason for recommending a major review is the lack of focus in the paper and also some paragraphs are difficult to understand. I suggest to review what precise questions you are asking of the data, clearly formulate hypotheses based on previous research and stick with these tightly throughout the paper. Points are made in the introduction but especially the discussion that are not directly related to the questions asked in this paper and can be shortened or removed.

Overall it is a good paper and after some changes will be ready for publication.

If you have any questions or would like me to specify further on my comments, please don't hesitate to get in contact.

Kind Regards,

David Ó' Reilly

Reviewer #2: 1. General – language. For publication in PLOS ONE, the article should be “presented in an intelligible fashion and is written in standard English”. The authors need to spellcheck and go through their manuscript. Getting language editing help is recommended. Several errors exist, some examples:

a. Table 1 “fellers” should be “fallers”

b. Line 129, word “balanceassessmenttt”

c. Line 176, sentence “The numbers the falls did not show any correlations”

d. Line 192 is missing punctuation.

e. Line 218 “what might suggest to be less valuable than FR test.”

Also, when abbreviating, make sure you do not have to abbreviate again. See for example sentence on line 141-142: “…it was named as a functional forward stability indicator (FFSI). The calculation of the forward functional stability indicator [%] (FFSI) was conducted…”

Introduction

2. Why is there no citation after “The ES experience falls due to significant deterioration of balance ability associated with decrease of muscle strength, sensory function and widespread degeneration in the central nervous system.”?

3. Please revise the unclear sentence “The questionnaire refers to the most of the basic, everyday activities presented by disabled people would be likely to have difficulty with because of their FOF.”

4. Limit of Stability is not abbreviated at the first time of use.

5. “The calculation of maximum forward lean range with respect to the mechanical (anatomical) stability boundary should allow to demonstrate the real safety margin. It is crucial for appropriate assessment of fear and risk of falling.” – Says who? Says you? Nothing in this sentence supports just using forward lean range. Medio-lateral leaning and LOS could be meaningful as well! Please revise and use references when building argumentation. Also, you can never be sure that you reach the “real” safety margin, as participants may stop themselves from leaning as far as they possibly can out of fear of falling or reduced joint flexibility. It is very difficult to achieve the “real” safety margin. I suggest you rephrase to something like “ should allow us to reach the boundary that is in close proximity to the real safety margin”

6. “Since falls usually occur during movements where the COG is shifted near or outside BOS, it seems that the LOS test, is the most appropriate measurement.” – Again, no references to support this statement. Also if falls usually occur during movements, is this really associated with a LOS test where a participants stands bilaterally, relatively static, and leans in different directions? Shouldn’t it rather be associated with people walking, stepping and how the COG moves within the LOS then?

Methods

1. The recruitment and inclusion procedure is not described thoroughly. It says that sixteen PD subjects and sixteen age-matched healthy controls voluntarily participated. Was this a convenience sample? How did recruitment proceed? Was this done at a hospital? Were the controls matched only on age? How will eventual non-matched variables confound your results?

2. Please indicate how many men and women there were.

3. “The research was accepted by the Institutional Bioethics Committee.” – where is this committee situated? More details required.

4. Table 1 – p-values are given but can be misinformative when there is risk for low statistical power. I encourage the authors to present confidence intervals instead. Also, the statistic section says that normal distribution was controlled for, but there is no information whether the parameters were normally distributed or not. Also, the “independent-samples t-test was conducted”. Did you use a parametric or non-parametric t-test?

5. Table 1 – I suggest putting percentages after the values on low-moderate-high concern to make it clearer.

6. Table 1 – for BBS, FR, Tinetti test and TUG; indicate what unit is presented (score, seconds, etc.), e.g. “TUG (s)”

7. From the methods it becomes clear that the study investigates fall history and not prospective falls. This should be indicated in the manuscript title as “history of falls”. This means that the outcome is prevalent before the risk factor is measured, and could likely influence the results. This should be mentioned in the limitation section.

8. The forward leaning LOS test is used. The advantages and disadvantages of this test needs to be discussed more. What information on LOS is missed? The authors early cite Forth et al. 2011 that in their paper states “However, an individual’s functional stability boundary during bipedal stance is more likely an ellipse much smaller than the base of support” The forward lean LOS test does not cover the other directions in an ellipse, and falls can occur in several directions. If the manuscript should be placed in the context of fall prediction, the validity of the test, e.g. strength and weaknesses should be properly discussed.

9. “The LOS test lasted 30 seconds and was repeated three times.” – were all three values analyzed? Or did you use the maximal value? Or an average over the three? Please specify. If using an average – discuss if you think test fatigue might influence the results.

10. I suggest helping the reader understand the FFSI better by adding this parameter its unit of measure (%?) to Table 1

Results

1. Table 2 – what does the red numbers indicate? Please specify

2. Table 3 – it is said to report “number and direction of falls”. Please specify, did you record in what direction the participants had fallen? And if so, this is not reported in table 3.

Discussion

1. Line 235-236 “Computerized posturography should be considered a more appropriate measure of balance ability among healthy ES than subjective tests.” What about the fact that most objective postural stability measures are performed in a static stance, while clinical tests are more dynamic in their nature? And falls often occur during movement and everyday situations, and not necessarily when standing still – for example, see Robinovitch et al. 2013 in the Lancet that have video-recorded falls.

2. Line 254-255 “However, an identification of postural control changes that are invisible to a naked eye should be pursued.”

Line 258-260 “We can conclude that based our results, it is not necessary to use the whole battery of clinical tests for balance assessment and still achieve satisfactory diagnostic outcomes.”

These statements require a better rationale. Why should we use objective measures? For fall risk? Can you really be sure this is better? There is evidence that self-reported fall history is a better predictor of prospective falls than objective tests, see for example Johansson et al. 2019 in Human Movement Science. So why should clinics spend resources on expensive equipment when it is better to ask a simple question?

3. Conclusion on line 300-301 states “Functional forward stability indicator is a good predictor of fear of falling in group of elderly people.” – The manuscript should change its title to reflect this conclusion, so that it is about predictors of fear of falling rather than actual falls. The reader might misinterpret this and think that prospective falls are investigated.

4. Line 224 “Existence of safety margin indicates deficits in balance.” – This single-sentence statement is unclear to me. Every human should have a safety margin, the question is how close to the safety margin our COG is displaced, and in turn might indicate deficits in balance.

5. Also Line 224 “The greater safety margin is combined with greater postural instability (10).” If by safety margin you mean the margin between COG and LOS, this is not true for falls. Again, see Johansson et al. 2019 in Human Movement Science. Here, the greater the area of the center of pressure in relation to the LOS, the greater the risk for incident falls. This means that a greater safety margin was beneficial and reduced the fall risk.

6. Line 281 – please be consequent with your citation method. Here you have switched to writing out the names and the year, and have not used the numbered reference system used in the rest of the manuscript.

6. PLOS authors have the option to publish the peer review history of their article (what does this mean?). If published, this will include your full peer review and any attached files.

Reviewer #1: Yes: David Ó' Reilly

Reviewer #2: No

---

## [Author Response · Author response to Decision Letter 0]

20 Jun 2020

Editor comment:

Answer: 

In the current revision we have followed all the PLOS ONE style requirements including those for file naming.

Answer: 

The additional details regarding participant consent were provided. The following sentences were added to the methods section: “The subjects gave informed written consent for voluntary participation in the study. The research was approved by the Institutional Ethics Committee of the Medical University of Warsaw (number KB/28/2014).”

3. Please ensure that you refer to Figure 1 in your text as, if accepted, production will need this reference to link the reader to the figure.

We have reformatted the equation in Word’s “Equation Tools” and removed equation as a figure form current manuscript. It is now referred to as an equation.

4. We note you have included a table to which you do not refer in the text of your manuscript. Please ensure that you refer to Table 1 in your text; if accepted, production will need this reference to link the reader to the Table.

Answer: 

The reference in the text to Table 1 was added: …“Sixteen PD subjects and sixteen anthropometrically matched healthy controls voluntarily participated in the study (Table 1).”

5. Please include captions for your Supporting Information files at the end of your manuscript, and update any in-text citations to match accordingly. 

We have added Supporting Information files at the end of the manuscript.

 

Reviewer #1

Dear authors, Thank you for your submission and your contribution to the field. This study is interesting with some worthy outcomes. I have attached a copy of your submission with some comments and suggestions. In summary, my main reason for recommending a major review is the lack of focus in the paper and also some paragraphs are difficult to understand. I suggest to review what precise questions you are asking of the data, clearly formulate hypotheses based on previous research and stick with these tightly throughout the paper. Points are made in the introduction but especially the discussion that are not directly related to the questions asked in this paper and can be shortened or removed. Overall it is a good paper and after some changes will be ready for publication.

Answer:

First of all, we would like to thank the reviewer for a very insightful review. The manuscript was amended thoroughly according to the reviewer’s comments and suggestions. Some issues have risen after reading the review, which is explained in detail below. Referring to the reviewer's concerns about language, we have sent our manuscript to The American Journal Experts (the certificate was attached). So all the proofread suggestions have been done.

Reviewer comment

The paragraph (lines 55-59/after correction lines 55-59) is not very clear to me..I understand that increased fear of falling may be a reaction to physical/cognitive deteriorations that can lead to beneficial cautious behaviour but it can also exacerbate the risk of falling.

Answer:

To clarify this sentence we have provided changes which are the following: 

Lines 51-56: “The fear of falling (FOF) could appear well before and/or after an incident of fall which creates a vicious cycle. Concomitant psychological symptom of falls is the fear of falling, which is common among older adults whether or not they have sustained a fall. FOF might also be a proper reaction to certain situations, leading the elderly to be cautious behaviour, and can contribute to fall prevention through careful choice of movement activity. Nevertheless, ESs increasingly cease physical activity because of the FOF (5).”

Reviewer comment

Lines 80-86: This problem statement needs to be made clearer .I believe you wish to compliment the disadvantages of clinical measures with more objective measures of stability however this is not directly stated.

Answer:

We have added following sentence to clarify the problem:” Clinical tests are often based on a qualitative assessment (BBS, Tinetti test). Despite the clear assessment instructions, the researcher may influence the test results.”

Reviewer comment

Your aim is not specifically addressing the problem statement in the previous paragraph… too general…perhaps with the aim of finding objective predictors of fall risk that can support/replace clinical assessments while also being specific to the patient needs is what you are aiming at? ..sample entro-py is also not described.

Answer:

We agree with the reviewer, that the aim of our study is too general, so we have changed this paragraph as follows: 

Lines: 101- 111 - “The aim of this study was to determine an objective predictor of the fear of falling and falls in ESs and patients with neurodegenerative disorders. For this purpose, the clinical and laboratory tests were analyzed. Depending on the study groups, the proposed objective measure could replace or support clinical assessment. Posturographic examination can also register early and nonvisible postural control changes, so we assumed that this objective assesment might replace clinical tests among ESs. We also hypothesized that the identification of the real functional limit of support with the LOS test would allow us to assess not only the fear of falling but also the risk of falls. Neurological disorders lead to changes in the mechanism of postural control (4). We assumed that PD patients present different balance strategies when their COG is located near their stability boundary. The analysis of the sample entropy should allow us to identify these changes.” 

We believe that these corrections would facilitate reading and understanding the discussion. 

Reviewer comment

Include number of males and females in each group.

Answer:

This information has been added in table 1. There were 9 males and 7 females in the PD group, and 8 males and 8 females in control group. 

Reviewer comment

How did you ensure there was no significant cognitive impairment in the PD sample?

Answer:

PD subjects had the results of the Mini Mental State Examination (MMSE) in their medical history. All of them obtained more than 26 points. We have added this information in eligibility criteria as follows: “… no consent to participate in the research; dementia or cognitive impairment (MMSE score below 26 points); and neuromuscular, vestibular or orthopedic disorders “

Reviewer comment

Summary statistics for all measurements should be presented, whether they are significant or not…not just sample entropy.

Answer:

All results have been presented in the manuscript – significant or not. The reviewer might have not noticed this fact because in order not to duplicate the data presentation the results of the comparison between groups are located in the main text of the manuscript (FFSI, sampEn) and in table 1 (FES-I, BBS, FR, Tinetti test, TUG). We described the results of FFSI and SampEn in details, because we built our conclusion based on these observations. All correlations are presented in table 2 and 3. 

Reviewer comment

A summary of the findings in your study and if they met or did not meet your hypotheses would be good to start with.

Answer:

Thank you for this comment. Following this suggestion we have pointed our hypothesis in our discussion as follows:

1) Line 256-257: “…So the existence of a safety margin might indicate deficits in balance, thus our research hypothesis was validated…”

2) Line 292-295: “…We can conclude that, based on our results, it is not necessary to use the whole battery of clinical tests for balance assessment to achieve satisfactory diagnostic outcomes, and our research hypothesis was validated.…”

3) Line 304-306 “…Our results validated our research hypothesis; therefore we postulate that balance assessments of the PD population should be made both with clinical tests and with computerized posturography …”

4) Line 326-327: “…Therefore, we confirmed our research hypothesis that the LOS test is a more challenging motor task for PD patients …”

Reviewer comment

This paragraph (221-233/after correction 252-266) needs to be made clearer… there are many points being made that are not aligned, making it difficult to read.

Answer: 

Thank you for pointing this out. We have revised this paragraph and made all necessary grammar and spelling corrections. We believe that it is now more clear. We have also edited and added following sentences: “A greater safety margin decreases the range of the maximum forward lean, thereby decreasing functional movement. So the existence of a safety margin might indicate deficits in balance…”

Reviewer comment 

Figure 3 should be in the methods in my opinion

Answer:

We appreciate this comment, however figure 3 (after correction figure 2) already presents some of results , therefore we put it in the discussion section.

Reviewer comment 

Line: 239-252(after corrections lines:272-282) Much of this can be summarized in a smaller number of sentences

Thank you for this comment. The paragraph was revised and made more concise for the sake of clarity and readability. The number of sentences was reduced.

Reviewer comment 

Line: 265-267 (after corrections lines 279-282)“Therefore, the LOS test being less prone to such strategy alterations during the performance, is in our opinion, more appropriate to early identification of balance deficits” – evidenced or opinion?

Answer:

We fully understand reviewer’s concern. We have changed our hypothesis as follow: Depending on the study groups, the proposed objective measure could replace or support clinical assessment. Posturographic examination can also register early and nonvisible postural control changes, so we assumed that this objective assessment might replace clinical tests among ESs.” Therefore, we have presented our opinion based on the results.

Reviewer comment

Line 277-280 (after correction lines 292-296) “We can conclude that based our results, it is not necessary to use the whole battery of clinical tests for balance assessment and still achieve satisfactory diagnostic outcomes. Acquiring more information in less time is especially important when conducting diagnosis in specific populations and situations.” – was this an aim?

Answer:

Once again we fully understand reviewer’s concern. We have changed our hypothesis, so statement in the current form correspond to the aim of our study.

Reviewer #2

Introduction

Reviewer comment 1:

General – language. For publication in PLOS ONE, the article should be “presented in an intelligible fashion and is written in standard English”. The authors need to spell check and go through their manuscript. Getting language editing help is recommended.

Answer:

At the beginning, we would like to express thanks to the reviewer for a very insightful review. The manuscript was corrected according to the reviewer’s suggestions. Your comments have contributed to improvement our paper. The manuscript was proofread by a certified proofread agency (certificate attached in the files).

Reviewer comment 2:

Why is there no citation after “The ES experience falls due to significant deterioration of balance ability associated with decrease of muscle strength, sensory function and widespread degeneration in the central nervous system.”?

Answer:

Thank you for pointing this out. We have added missing citation: Ambrose AF, Paul G, Hausdorff JM. Risk factors for falls among older adults: A review of the literature. Maturitas.2013; 75 (1): 51-61.

Reviewer comment 3

Please revise the unclear sentence “The questionnaire refers to the most of the basic, everyday activities presented by disabled people would be likely to have difficulty with because of their FOF. :

Answer:

We have revised this sentence:“The questionnaire refers to the most of the basic, everyday activities. Only people with FOF may have difficulty to perform these actions.”

Reviewer comment 4

Limit of Stability is not abbreviated at the first time of use.

Answer:

Thank you for pointing this out. We have abbreviated Limit of Stability (LOS) at the first time of use .

Reviewer comment 5

The calculation of maximum forward lean range with respect to the mechanical (anatomical) stability boundary should allow to demonstrate the real safety margin. It is crucial for appropriate assessment of fear and risk of falling.” – Says who? Says you? Nothing in this sentence supports just using forward lean range. Medio-lateral leaning and LOS could be meaningful as well! Please revise and use references when building argumentation. Also, you can never be sure that you reach the “real” safety margin, as participants may stop themselves from leaning as far as they possibly can out of fear of falling or reduced joint flexibility. It is very difficult to achieve the “real” safety margin. I suggest you rephrase to something like “ should allow us to reach the boundary that is in close proximity to the real safety margin”

Answer:

Thank you for rising this issue. To clarify why we focused our research only on the forward stability boundary, we have added the following sentences to the introduction: “In older adults over 65 yr, 60% of falls occur in a forward direction (O’Neill et al. 1994). Additionally, Youn et al. (2017) noticed that more than 70% of PD patients had a history of falls in the forward direction. Therefore, in the present study, we focused on exploring the anterior stability boundary”. We also agree with the reviewer, that we can never be sure that we reach the “real” safety margin, as participants may stop themselves from leaning as far as they possibly can out of the fear of falling or reduced joint flexibility. Therefore, we have rephrased this sentence according to the reviewer’s suggestion: “The calculation of the maximum forward lean range with respect to the mechanical (anatomical) stability boundary should allow subjects to reach the boundary that is in close proximity to the real safety margin”. We also have added references: Błaszczyk J.W., Lowe D. Ranges of postural stability and their changes in the elderly. Gait & Posture 1994(2):11-7. The authors have pointed out the importance of the detection of safety margin: “Reduction of the maximum voluntary excursion increases the probability of recovery from postural instability by increasing the margin safety, which allows for more time to complete the recovery programme”.

Reviewer comment 6

“Since falls usually occur during movements where the COG is shifted near or outside BOS, it seems that the LOS test, is the most appropriate measurement.” – Again, no references to support this statement. Also if falls usually occur during movements, is this really associated with a LOS test where a participants stands bilaterally, relatively static, and leans in different directions? Shouldn’t it rather be associated with people walking, stepping and how the COG moves within the LOS then?

Answer:

We agree that this statement should be supported by proper reference. However, this statement was created based on knowledge about postural strategy. Upright posture is controlled during disturbance by the ankle and hip joint movements, classified as the “ankle strategy” and “hip strategy”, respectively. The greater the postural disturbance, the closer the COG moves to the stability boundary. When the COG crosses the stability boundary, the subject has to take a step to recover the upright posture. According to Clark et al. (2005), for many older adults, spatial and temporal control of the COG is decreased, and movement strategies become unwieldy, resulting in an increased risk of falls when performing reaching tasks or gait initiation. Therefore, we have rephrased the sentence as follows: “With an inefficient postural strategy (ankle and hip strategy) in elderly individuals, falls usually occur during movements where the COG is shifted near or outside the BOS. It seems that the LOS test is the most appropriate measurement.” We have also added the appropriate reference (Cleworth et al. 2018 Gait & Posture).

Methods:

Reviewer comment 1

The recruitment and inclusion procedure is not described thoroughly. It says that sixteen PD subjects and sixteen age-matched healthy controls voluntarily participated. Was this a convenience sample? How did recruitment proceed? Was this done at a hospital? Were the controls matched only on age? How will eventual non-matched variables confound your results?

Answer: 

Again, thank you for pointing this out. We have provided several corrections in the description of this section. We recruited PD subjects from the Local Association of Parkinson's disease community. We have changed the phrase from “PD patients” on “PD subjects”. One of our co-authors, who is the specialist of PD physiotherapy verified the inclusion and exclusion criteria. Subjects who met eligibility criteria were invited to examination. The research took place at the Academy of Physical Education in Katowice (Poland). The sample size of PD subjects was determined by the number of subjects from the Association of Parkinson's Disease who met the eligibility criteria. The control group was matched not only for age, but also for body height and weight. Table 1 presents the anthropometric values of the two groups. All of the above details have been included in the manuscript.

Reviewer comment 2

Please indicate how many men and women there were

Answer:

Thank you for this comment. This information was added to the manuscript (table 1). In the PD group there were 9 males and 7 females, in control group there were 8 males and 8 females. 

Reviewer comment 3

The research was accepted by the Institutional Bioethics Committee.” – Where is this committee situated? More details required. 

Answer:

We have changed the sentence as follow: “The research was accepted by the Institutional Ethics Committee Medical University in Warsaw number KB/28/2014”.

Reviewer comment 4

Table 1 – p-values are given but can be misinformative when there is risk for low statistical power. I encourage the authors to present confidence intervals instead. Also, the statistic section says that normal distribution was controlled for, but there is no information whether the parameters were normally distributed or not. Also, the “independent-samples t-test was conducted”. Did you use a parametric or non-parametric t-test?

Answer:

We agree that the section of statistical methods should be written with more details. Depending on the variable distribution, parametric or nonparametric statistical analysis was used. All anthropometric variables, the FFSI and the SampEn, were normally distributed compared to FES-I, BBS, TUG, FR and Tinetti test scores; therefore, we used parametric and nonparametric tests, respectively. To compare values of age, weight, body weight, the FFSI and the SampEn between groups, an independent-sample t-test was conducted. To compare FES-I, BBS, TUG, FR and Tinetti test scores, the Mann-Whitney U test was conducted. All correlations were calculated by the nonparametric Spearman rank test. We have corrected this section as follows: “The parametric and nonparametric tests were used. An independent-sample t-test was conducted to compare the anthropometric variables FFSI and Samp En from ESs and PD patients. The Mann-Whitney U test was conducted to compare FES-I, BBS, Tinetti, FR and TUG test scores from ESs and PD patients.” We have removed p-values from table one and we have added confidence intervals.

Reviewer comment 5

Table 1 – I suggest putting percentages after the values on low-moderate-high concern to make it clearer.

Answer:

We have put percentages after the values on low-moderate-high concern in table 1.

Reviewer comment 6

Table 1 – for BBS, FR, Tinetti test and TUG; indicate what unit is presented (score, seconds, etc.), e.g. “TUG (s)”

Answer:

Table 1 was revised according to reviewers comments. 

Reviewer comment 7

From the methods it becomes clear that the study investigates fall history and not prospective falls. This should be indicated in the manuscript title as “history of falls”. This means that the outcome is prevalent before the risk factor is measured, and could likely influence the results. This should be mentioned in the limitation section. 

Answer:

Once again, thank you for pointing this out. We have changed the title for: ” Fall-related measures in elderly individuals and Parkinson’s disease subjects”. We also added to the limitation section following sentences: “Future research should be focused on prospective risk of falls. Future research should be focused on the prospective risk of falls. It will be useful to create FFSI norms that can help diagnose the risk of falls.”

Reviewer comment 8

The forward leaning LOS test is used. The advantages and disadvantages of this test needs to be discussed more. What information on LOS is missed? The authors early cite Forth et al. 2011 that in their paper states “However, an individual’s functional stability boundary during bipedal stance is more likely an ellipse much smaller than the base of support” The forward lean LOS test does not cover the other directions in an ellipse, and falls can occur in several directions. If the manuscript should be placed in the context of fall prediction, the validity of the test, e.g. strength and weaknesses should be properly discussed.

Answer:

Thank you for pointing this out, as we have mentioned before, we focused on forward stability boundary, because of the frequency of falls in this direction. We wrote about some advantages and disadvantage of the LOS test in section of discussion.

Reviewer comment 9

“The LOS test lasted 30 seconds and was repeated three times.” – were all three values analyzed? Or did you use the maximal value? Or an average over the three? Please specify. If using an average – discuss if you think test fatigue might influence the results. 

Answer:

Thank you for this comment. We have added calculation of the intraclass correlation coefficients for the limit of stability test in the statistical section as follows: “The levels of correlation were considered weak positive (0 – 0,3), weak negative (-0,3 – 0), moderate positive (0,3 – 0,7), moderate negative (-0,3 – (-0,7)), high positive (0,7 – 1,0), and high negative (-0,7 – (-1,0)) (18). Intraclass correlation coefficients (ICCs) were calculated for the LOS test (FFSI, sampEn phase 1, sampEn phase 3). The levels of reliability were considered poor (ICC < 0.50), moderate (0.50 ≤ ICC < 0.75), good (0.75 ≤ ICC < 0.90), and excellent (ICC ≥ 0.90), according to Portney et al (2000). The LOS test achieved excellent reliability with 3 repetitions..” – this of course means that we took average of three trials for further analysis. The fatigue probably might influence the result of the test if there were not rest break between repetitions. That is why each research subjects were getting off the force plate after each trial and could rest.

Reviewer comment 10

I suggest helping the reader understand the FFSI better by adding this parameter its unit of measure (%?) to Table 1

Answer:

Thank you for this suggestion. We have added measurement units of FFSI to the Table 1.

Results

Reviewer comment 1

Table 2 – what does the red numbers indicate? Please specify

Answer:

Again, thank you for the comment. Indeed, the red color might be misleading. So we have changed red to bold indicating statistical significance. The description of this was added to the table’s legend. 

Reviewer comment 2

Table 3 – it is said to report “number and direction of falls”. Please specify, did you record in what direction the participants had fallen? And if so, this is not reported in table 3.

Answer:

Thank you for pointing this out. It was a mistake. We didn’t examine the direction of the falls. So we have corrected the title of Table 3 as follows: “Spearman’s correlation coefficients between among balance clinical test scores, the FES-I score, and the number of falls in elderly subjects and PD subjects.

Discussion 

Reviewer comment 1 

Line 235-236 “Computerized posturography should be considered a more appropriate measure of balance ability among healthy ES than subjective tests.” What about the fact that most objective postural stability measures are performed in a static stance, while clinical tests are more dynamic in their nature? And falls often occur during movement and everyday situations, and not necessarily when standing still – for example, see Robinovitch et al. 2013 in the Lancet that have video-recorded falls.

Answer:

We fully understand the reviewer’s concern. We agree that most objective measures are performed in a static stance, but falls predominantly occur during more dynamic motor tasks. Therefore, we chose the limit of stability test rather than quiet standing. Almost 30 years ago, Duncan et al. (1990) created a simple measure such as the Functional Reach Test, which was highly correlated with the risk of falls. Over the years, it turns out that indications of the functional limit of stability are crucial to assess the risk of falls. Even simple motor tasks such as body leaning on the force plate are very informative. We know exactly where the COP is located within the base of support. This allows for the calculation of the safety margin. No clinical test allows this. Additionally, having a physiotherapist’s experience, one can notice that the researcher has a large influence on the BBS and Tinetti test results.

Reviewer comment 2 

Line: 254-255 “However, an identification of postural control changes that are invisible to a naked eye should be pursued.”

Line 258-260 “We can conclude that based our results, it is not necessary to use the whole battery of clinical tests for balance assessment and still achieve satisfactory diagnostic outcomes.”

These statements require a better rationale. Why should we use objective measures? For fall risk? Can you really be sure this is better? There is evidence that self-reported fall history is a better predictor of prospective falls than objective tests, see for example Johansson et al. 2019 in Human Movement Science. So why should clinics spend resources on expensive equipment when it is better to ask a simple question? 

Answer:

We fully understand reviewer’s concern, however self-reported fall history assesses what happened in the past. We agree that, based on this report, research subjects have poor balance. However, is it possible to predict a fall before the first accident? The fear of falling can appear both before and after accidents or falls. In our study, we detected the fear of falling without a history of falls in almost all research subjects. Therefore, our indicator has great potential to determine the fear of falling. There is a high probability of detecting the risk of falling if the research is conducted with follow-up examinations. Thank you for suggesting Johansson et al. papers, we have added it to references

Reviewer comment 3

Line: Conclusion on line 300-301 states “Functional forward stability indicator is a good predictor of fear of falling in group of elderly people.” – The manuscript should change its title to reflect this conclusion, so that it is about predictors of fear of falling rather than actual falls. The reader might misinterpret this and think that prospective falls are investigated 

Answer:

We completely agree with reviewer comment. So as we have mentioned before, we have changed the title of the manuscript as follows: “Fall related measures in elderly and Parkinson’s disease”. 

Reviewer comment 4

Line: Conclusion Line 224 “Existence of safety margin indicates deficits in balance.” – This single-sentence statement is unclear to me. Every human should have a safety margin, the question is how close to the safety margin our COG is displaced, and in turn might indicate deficits in balance.

Answer:

We do agree with this logic. However, we wanted to present other ideas connected with functional efficiency. We agree that a greater safety margin indicates more time to introduce and choose the strategy of balance. However, a greater safety margin decreases the range of the maximum forward lean, thereby decreasing functional movement. All subjects should explore as much as possible of the base of support. In our understanding, the safety margin is located between the first metatarsal phalangeal joint and the location of the COP while keeping the body in a leaning position. If the COP is located closer to this joint, the safety margin has a smaller area. We hope that the reviewer would agree with this rationale. 

Reviewer comment 3

Also Line 224 “The greater safety margin is combined with greater postural instability (10).” If by safety margin you mean the margin between COG and LOS, this is not true for falls. Again, see Johansson et al. 2019 in Human Movement Science. Here, the greater the area of the center of pressure in relation to the LOS, the greater the risk for incident falls. This means that a greater safety margin was beneficial and reduced the fall risk.

Answer:

In our study, smaller values of the range of maximum forward lean correlated with lower scores on some clinical tests (the BBS, FR test, and Tinetti test) and with higher scores on the TUG test. This clearly means that lower FFSI values indicate a greater risk of falling. Furthermore, a smaller range of maximum forward lean results in a greater safety margin

 Reviewer comment 3

Line 281 – please be consequent with your citation method. Here you have switched to writing out the names and the year, and have not used the numbered reference system used in the rest of the manuscript.

Answer:

Thank you for pointing this out. All citations have been checked once again and we have made relevant correction.

---

## [Decision Letter · Decision Letter 1]

26 Jun 2020

PONE-D-20-08972R1

Fall-related measures in elderly individuals and Parkinson’s disease subjects

PLOS ONE

Dear Dr. Michalska,

Thank you for submitting your manuscript to PLOS ONE. After careful consideration, we feel that it has merit but does not fully meet PLOS ONE’s publication criteria as it currently stands. Therefore, we invite you to submit a revised version of the manuscript that addresses the points raised during the review process.

The two original reviewers carefully reevaluated the revision. Their comments are appended below. One referee is satisfied with this revision, while the other reviewer still has some concerns regarding the way of statistical analysis and the composition of Introduction and Discussion part. I would suggest to make revision according to these criticisms for better understanding of your manuscript.

We look forward to receiving your revised manuscript.

Kind regards,

Manabu Sakakibara, Ph.D.

Academic Editor

PLOS ONE

Reviewers' comments:

Reviewer's Responses to Questions

**Comments to the Author**

1. If the authors have adequately addressed your comments raised in a previous round of review and you feel that this manuscript is now acceptable for publication, you may indicate that here to bypass the “Comments to the Author” section, enter your conflict of interest statement in the “Confidential to Editor” section, and submit your "Accept" recommendation.

Reviewer #1: (No Response)

Reviewer #2: All comments have been addressed

2. Is the manuscript technically sound, and do the data support the conclusions?

Reviewer #1: Partly

Reviewer #2: Yes

3. Has the statistical analysis been performed appropriately and rigorously? 

Reviewer #1: No

Reviewer #2: Yes

4. Have the authors made all data underlying the findings in their manuscript fully available?

Reviewer #1: Yes

Reviewer #2: Yes

5. Is the manuscript presented in an intelligible fashion and written in standard English?

Reviewer #1: Yes

Reviewer #2: Yes

6. Review Comments to the Author

Reviewer #1: Dear Authors,

Thank you for re-submitting a revised version of the manuscript. I see you have went to good efforts to meet the comments made in the last revision. I have attached the revised document with some further comments. I have some concerns regarding the statistical analysis that have come to light in this review as the paper is now clearer to me. I believe a bonferroni correction is required for p-values reported that reflects the number of comparisons made in this study. This will reduce the risk of type 1 error but will inevitably make some of your results insignificant.

I therefore suggest that one either conducts a bonferroni correction or makes their statistical analysis more concise with fewer comparisons while still maintaining their aims...I also suggest that the intro to the discussion section be revised as this section is quite difficult to read without a proper summary of results and what the hypotheses were. It is important to address why you are making the points you are making, what hypotheses they address in your study etc. as the reader may need to go back and forth between sections otherwise which isn't optimal. Appropriate referencing is something that needs to be looked over during this revision also as some paragraphs are without them or require more.

After this is satisfactorily carried out i would be happy to recommend for publication as the findings are quite interesting.

Kind Regards,

David Ó' Reilly

Reviewer #2: The authors’ responses to my comments and questions are satisfactory. The rationale why they chose to investigate maximum forward lean (MFL) as a predictor for fear of falling is now clearer. LOS is a complex measure depending on how the postural stability within the stability boundary is assessed, and the authors have provided argumentation why the MFL can be relevant. Overall, the manuscript is clearer and linguistically improved, enhancing its readability.

I have one additional comment that requires attention. It is not as severe as needing a 2nd round of review and can perhaps be checked by the editor:

Table 1 – confidence intervals for BBS (score) overlap, yet there is a bold marking for statistical significance. Please adjust accordingly, if it was significant and this is a typo or not.

7. PLOS authors have the option to publish the peer review history of their article (what does this mean?). If published, this will include your full peer review and any attached files.

Reviewer #1: **Yes: **David Ó'Reilly

Reviewer #2: No

---

## [Author Response · Author response to Decision Letter 1]

14 Jul 2020

Reviewer #1

Reviewer comment

References required in this paragraph (line: 50-54).

Answer:

Two references were added to support this paragraph: “Park et al. (2014) Am J Phys Med Rehabil.” and “Young et al. (2015) Gait Posture.”

Reviewer comment

“Neurological disorders lead to changes in the mechanism of postural control.”- this statements requiring a reference should not be included in the aims paragraph as they should have already been stated previously and the aims and hypotheses should be self-evident.

Answer:

Thank you for this comment. Following this suggestion we have removed this statement. We agree that statements requiring a reference should not be included in the aims paragraph. We wrote about the changes in the mechanism of postural control in PD subjects in the third paragraph of the introduction. 

Reviewer comment

Table 1 – Was a t-test or Mann Whitney U used? If so report the values so replication can be made possible.

Answer:

Table 1 was revised. Depending on the distribution of variables, the values of t-test or Mann-Whitney U test were added to the table 1. 

Reviewer comment

Equation label required to differentiate it from figures 

Answer:

The equation label was added and incorporated in the text of the manuscript.

Reviewer comment

I believe multiple comparisons requires a bonferroni correction to reduce type 1 error risk…for example by simply dividing 0.05 by the number of tests conducted for each hypotheses, one can arrive at the appropriate p value for significance…here it is 0.05 divided by 6 (the number of balance measures compared against the FFSI) giving you a p-value of 0.00833.

Answer:

Thank you for raising this issue. The significance level had been revised and the Bonferroni correction was implemented (p = 0.0083) to reduce the possibility of statistical error. The information was added in the section of statistical:

 „The Bonferroni correction was used to decrease risk of a type I error during making multiple statistical tests”. 

We have also changed the indication of statistically significant on table 2. 

Reviewer comment

I believe Similarly, a Bonferroni correction will be required here also to reduce the risk of type 1 error..in this case it will need to be 0.05 divided by 8 as there is two dependent variables and two hypotheses giving you a p-value of 0.00625 for PD and ES subject comparisons..if you would like to avoid this correction it is best to reduce your analysis to an evidenced based search for specific correlations that are meaningful towards your overall aim..this could be carried out by selecting a smaller number of well-established balance assessments to compare against FFSI that address aspects of balance control individually….

Answer: 

Once again, thank you for this comment. As above, the significance level had been revised with a Bonferroni correction (p = 0.00625) to reduce the possibility of statistical error. We have provided new statistical significance in table 3. 

Reviewer comment

This first paragraph doesn’t address your aims, the purpose of the study, summary of findings or really anything relevant to your study except stating how the PD subjects were qualified as an at risk group for falls and how the measures were not sensitive..the reader should be reminded and brought into each section with a progressive building towards the conclusion rather than point statements about the findings

Anwser:

Thank you for pointing this out. We have added a new paragraph in the discussion section as follows: 

„The main aim of this study was to determine an objective predictor of the fear of falling and falls in ESs and PD subjects. That is way, we have analyzed both the qualitative and quantitative balance diagnostic methods, indicating that the posturographic examination can both register early and nonvisible postural control changes. Additionally LOS test assesses safety margin, as an alert of fear of falling. First, we assumed that the identification of the real functional limit of support surface with the LOS test would allow to assess not only the fear of falling but also the risk of falls. Second, we hypothesized that depending on the study groups, the proposed objective measure could replace or support clinical assessment. Third, we supposed that PD subjects present different balance strategies when their COG is located near their stability boundary.” We have also provided all hypothesis in the main text for discussion.

We would like to thank again the reviewer for a very insightful review. Your comments have contributed to improvement our paper.

Reviewer #2 

Reviewer comment :

The authors' responses to my comments and questions are satisfactory. The rationale why they chose to investigate maximum forward lean (MFL) as a predictor for fear of falling is now clearer. LOS is a complex measure depending on how the postural stability within the stability boundary is assessed, and the authors have provided argumentation why the MFL can be relevant. Overall, the manuscript is clearer and linguistically improved, enhancing its readability. I have one additional comment that requires attention. It is not as severe as needing a 2nd round of review and can perhaps be checked by the editor:

Table 1 - confidence intervals for BBS (score) overlap, yet there is a bold marking for statistical significance. Please adjust accordingly, if it was significant and this is a typo or not. 

We are very grateful for a such an approving review. It strongly motivates us to do further research. There were no significant differences between ESs and PD subjects in BBS score. We have corrected a mistake in table 1.

---

## [Decision Letter · Decision Letter 2]

16 Jul 2020

Fall-related measures in elderly individuals and Parkinson’s disease subjects

PONE-D-20-08972R2

Dear Dr. Michalska,

We’re pleased to inform you that your manuscript has been judged scientifically suitable for publication and will be formally accepted for publication once it meets all outstanding technical requirements.

Kind regards,

Manabu Sakakibara, Ph.D.

Academic Editor

PLOS ONE

Additional Editor Comments (optional):

Reviewers' comments:

Reviewer's Responses to Questions

**Comments to the Author**

1. If the authors have adequately addressed your comments raised in a previous round of review and you feel that this manuscript is now acceptable for publication, you may indicate that here to bypass the “Comments to the Author” section, enter your conflict of interest statement in the “Confidential to Editor” section, and submit your "Accept" recommendation.

Reviewer #1: All comments have been addressed

2. Is the manuscript technically sound, and do the data support the conclusions?

Reviewer #1: Yes

3. Has the statistical analysis been performed appropriately and rigorously? 

Reviewer #1: Yes

4. Have the authors made all data underlying the findings in their manuscript fully available?

Reviewer #1: Yes

5. Is the manuscript presented in an intelligible fashion and written in standard English?

Reviewer #1: Yes

6. Review Comments to the Author

Reviewer #1: I am happy with the efforts made to ameliorate previous versions of this manuscript. The findings are relevant and informative for clinical assessment and balance evaluations. I can therefore recommend this manuscript for publication.

7. PLOS authors have the option to publish the peer review history of their article (what does this mean?). If published, this will include your full peer review and any attached files.

Reviewer #1: No

---

## [Editor Report · Acceptance letter]

23 Jul 2020

PONE-D-20-08972R2 

Fall-related measures in elderly individuals and Parkinson’s disease subjects 

Dear Dr. Michalska:

I'm pleased to inform you that your manuscript has been deemed suitable for publication in PLOS ONE. Congratulations! Your manuscript is now with our production department. 

Kind regards, 

on behalf of

Dr. Manabu Sakakibara 

Academic Editor

PLOS ONE